# Glutathione Modulates Efficacious Changes in the Immune Response against Tuberculosis

**DOI:** 10.3390/biomedicines11051340

**Published:** 2023-05-02

**Authors:** Arbi Abnousian, Joshua Vasquez, Kayvan Sasaninia, Melissa Kelley, Vishwanath Venketaraman

**Affiliations:** 1College of Osteopathic Medicine of the Pacific, Western University of Health Sciences, Pomona, CA 91766, USA; arbi.abnousian@westernu.edu (A.A.);; 2Graduate College of Biomedical Sciences, Western University of Health Sciences, Pomona, CA 91768, USA

**Keywords:** glutathione, tuberculosis, mycobacteria, interferons, interleukins, tumor necrosis factor, transforming growth factor

## Abstract

Glutathione (GSH) is an antioxidant in human cells that is utilized to prevent damage occurred by reactive oxygen species, free radicals, peroxides, lipid peroxides, and heavy metals. Due to its immunological role in tuberculosis (TB), GSH is hypothesized to play an important part in the immune response against *M. tb* infection. In fact, one of the hallmark structures of TB is granuloma formation, which involves many types of immune cells. T cells, specifically, are a major component and are involved in the release of cytokines and activation of macrophages. GSH also serves an important function in macrophages, natural killer cells, and T cells in modulating their activation, their metabolism, proper cytokine release, proper redox activity, and free radical levels. For patients with increased susceptibility, such as those with HIV and type 2 diabetes, the demand for higher GSH levels is increased. GSH acts as an important immunomodulatory antioxidant by stabilizing redox activity, shifting of cytokine profile toward Th1 type response, and enhancing T lymphocytes. This review compiles reports showing the benefits of GSH in improving the immune responses against *M. tb* infection and the use of GSH as an adjunctive therapy for TB.

## 1. Introduction

Glutathione (GSH) is a tripeptide and an antioxidant, which is mostly found in the cytosol, and synthesized through two ATP-requiring enzymatic steps. The first involves the formation of γ-glutamyl cysteine from glutamate and cysteine, followed by the formation of γ-L-glutamyl-L-cysteinyl-glycine from γ-glutamyl cysteine and glycine (Figure 1) [1]. The first step is the rate-limiting step and is catalyzed by glutamate–cysteine ligase (GCL) [1,2]. The second step is catalyzed by GSH synthase, which is subject to feedback inhibition by GSH itself [1,2]. In the presence of glutathione peroxidase, thiol-reduced glutathione (rGSH) reduces hydrogen peroxide formed from oxidative stress, resulting in the formation of oxidized glutathione (GSSG). Then, GSSG reductase (GSR) reduces GSSG back to rGSH using NADPH, forming a redox cycle [3]. Therefore, accumulation of GSSG and depleted levels of rGSH can increase oxidative stress, which is sometimes measured in experiments using levels of reactive oxygen species. However, in the case of severe oxidative stress, the ability of the cell to reduce GSSG to rGSH is diminished and leads to an accumulation of GSSG and depleted levels of rGSH [4]. Glutathione (GSH) is synthesized at high levels by cells during reactive oxygen intermediate and nitrogen intermediate production. GSH scavenges peroxide species and has been recently shown to play an important role in apoptosis and to regulate antigen-presenting cell functions [5,6]. Due to these immunomodulatory properties, GSH has been hypothesized to be part of an effective adjunctive therapy for *Mycobacterium tuberculosis* (*M. tb*) infection. In this review, we analyze the role of GSH supplementation in forming proper immune response against *M. tb* infection and its role in adjunctive therapy for tuberculosis (TB) patients.

The annual incidence rate for TB according to the World Health Organization is about 10 million [7]. Approximately 45% of the cases occurred in Southeast Asia, 23% in Africa, 18% in Western Pacific, 8.1% in the Eastern Mediterranean, 2.9% in the Americas, and 2.2% in Europe [7]. In some countries, the Bacille Calmette-Guérin (BCG) vaccine is given to prevent TB [8,9]. The vaccine prevents the disease outside of the lungs, but inside the lungs, it remains vulnerable. About a quarter of the world is thought to have been infected with *M. tb* and about 5–10% of that population acquire symptoms and develop TB [7]. Furthermore, the emergence of multidrug resistant strains of *M. tb*, along with immunocompromised individuals with HIV, have caused an increase in the incidence of TB. Among those with TB, about 6.7% have HIV and about 15% have diabetes [7,10]. In both conditions, individuals are susceptible to bacterial infections. HIV patients have lower T cell counts and weaker cell-mediated immunity, in addition to increased production of pro-inflammatory cytokines [11]. Patients with diabetes are also at a higher risk due to chronic inflammation caused by an increase in pro-inflammatory cytokines [12]. The excessive production of pro-inflammatory cytokines can lead to oxidative stress and the generation of free radicals. GSH scavenges these reactive oxygen species, and thus can be used as an adjunctive treatment option for these comorbidities.

## 2. Methods

TB remains a deadly infectious disease with the causative agent being *M. tb*. In this review article, our intention is to highlight the role GSH plays in the immune system in TB, especially in patients with HIV and type 2 diabetes. Our five co-authors carried out extensive research using two search engines and reviewed more than one hundred and ten articles initially. The literature obtained for this review was found using the following search engines: PubMed and Google Scholar. The terminology used for searching and finding articles included any combination of the following: “Glutathione”, “Tuberculosis”, “Mycobacteria”, “HIV”, “Type 2 Diabetes”, “free radicals”, “Interferons”, “Interleukins”, “Tumor necrosis factor”, “Transforming growth factor”, “Reactive oxygen species”, “T cells”, “enzymes”, “Natural killer cells”, “Macrophages”, and “N-acetyl cysteine”. Review articles were selected based on methods, significance, effect size, relevance, and credibility. Articles were excluded for non-relevance, non-significance, poor effect size, and old publication date. The manuscript was reviewed amongst the group members and the final version was edited in accordance with Dr. Venketaraman’s suggestions and advisory.

## 3. Granuloma and Cytokines

One of the hallmark structures of TB is granuloma formation. Granuloma, an aggregate of organized mature macrophages, separates TB from the surroundings. Upon infection, *M. tb* contact induces phagocytosis by resident alveolar macrophages. These mature macrophages have increased cytoplasmic size and more organelles, in addition to ruffled cell membranes that makes them more phagocytic [13]. Other immune cells that make up granuloma include neutrophils, dendritic cells, B and T cells, and natural killer cells. T cells, specifically, are a major component of granulomas and are important in containing *M. tb* infection [14,15]. They release cytokines necessary for initiation, and for activation of macrophages. Granulomas are an important component of the primary immune response to TB and individuals who do not show signs of active TB could have had healed or sterile granulomas [16,17].

Granulomas play a complex role in *M. tb* infection. Granuloma forms within the lungs and keeps TB inactive, which is referred to as latent TB [18,19]. Mycobacteria can also exploit granuloma formation, proliferate inside the structure, and disseminate throughout the body [13]. However, without granuloma formation, this dissemination would be greater. Therefore, local control of the infection is crucial. There are also many cases where TB infiltrates the lungs and escapes trafficking and immune response by the host. This is especially evident in immunocompromised patients, who have deficiency of tumor necrosis factor (TNF)-α, interferon (IFN)-γ, or interleukin (IL)-12 (Table 1) [13]. In immunocompromised individuals, GSH levels can also be significantly compromised [20]. Each granuloma is unique, even within the same host, and has a different proportion of different types of immune cells [14]. Furthermore, there are different proportions of cytokines in each granuloma. IL-2, IL-17, IFN-γ, and TNF-α all show an inverse correlation to bacterial burden [14]. T cell secretion of IFN-γ, TNF-α, and IL-17 is thought to be required for the activation of macrophages and inducing an immune response [14].

Cytokine balance is important for proper immune response against *M. tb*, and the proper ratio and proportion of the type of cytokines is crucial for the correct and efficient response against TB. In active TB, *M. tb*-specific T cells are suppressed, as shown by suppression of proper IL-2 and IFN-γ production, and overproduction of immunosuppressive cytokines, IL-10, and transforming growth factor (TGF)-β [21]. TGF-β can itself diminish GSH levels by downregulating the expression of enzyme subunits involved in GSH synthesis. Furthermore, in HIV positive patients, T-helper 1 (Th1) response is shifted to a T-helper 2 (Th2) response [22,23]. Therefore, mechanisms utilized to reverse these changes in cytokine profile are important, since T cells and proper Th1 type cytokine responses are a hallmark for the control of *M. tb* infection. Fortunately, in our previous studies, GSH supplementation resulted in an increase in Th1 cytokines, such as IL-2, IL-12, IFN-γ, and TNF-α [21,24].

In previous studies, we found that macrophages require both IFN-y and LPS, which is added to macrophages to result in TNF-α production, in order to have enhanced synthesis of inducible NO synthase (iNOS) and generation of nitric oxide (NO) [25]. NO is shown to have antimicrobial activity and inhibits the growth of *M. tb* [26,27]. We have also observed decreased levels of IL-10, TGF-β, and IL-6 by GSH supplementation [24]. IL-6 is a proinflammatory cytokine that promotes Th2 differentiation, inhibits Th1 response, induces oxidative stress, and causes systemic inflammation [28,29,30].

## 4. Role of GSH during Ferroptosis and in Macrophages

Cell death during *M. tb* infection is considered as detrimental since it allows the mycobacteria to spread. Ferroptosis is one type of cell death in which accumulation of iron and toxic lipid peroxides leads to the necrosis of the cell [31,32]. It was observed that macrophage necrosis was stimulated by reduced levels of GSH and glutathione peroxidase-4 (Gpx4), and under conditions of iron overload lipid peroxidation is triggered and if Gpx4 is suppressed, cell death presumes [31,33,34,35]. During usual conditions, lipid peroxides are reduced by Gpx4 through GSH oxidation, but when Gpx4 expression or activity is inhibited during iron overload, lipid peroxide levels lead to cell death [31,36].

GSH also plays other crucial roles in macrophages. Mycobacteria defective in the transport of peptides, such as GSH, are resistant to antimicrobial effects of GSH. In a study that examined the role of GSH in macrophages, and the intracellular growth of mycobacteria, mutant mice defective in the transport of GSH were found to be resistant to toxicity caused by GSH [25]. Furthermore, another study found that a virulent strain of *M. tb*, H37Rv, was sensitive to GSH and nitrosoglutathione (GSNO), in vitro [6]. To test the role of GSH in antimycobacterial activity, macrophages infected with H37Rv were treated with IFN-γ, LPS, and buthionine sulfoximine (BSO), and growth of H37Rv inside macrophages was measured. It should be noted that BSO inhibits the activity of the first step in the synthesis of GSH and can be used to differentiate antimycobacterial activity due to GSH levels. In one study, BSO-treated macrophages had a significant decrease in intracellular GSH levels and there was a significant increase in the intracellular growth of H37Rv [6]. However, the mechanism of action of GSH remains unclear, and it is hypothesized that high concentrations of GSH could result in an imbalance in redox activity within a bacterium containing alternative thiol responsible for the redox activity [6].

## 5. Natural Killer Cells

Natural killer (NK) cells are cytotoxic lymphocytes that are part of the innate immune system, which controls the spread of multiple types of microbial infections and tumors. They are also in the same family as T and B cells and similar to cytotoxic T cells, and release granules that contain enzymes, such as granulysin, that can kill infected cells by lysis [37,38]. They respond to virus-infected cells, cells infected with intracellular pathogens, and to tumor cells. Research has highlighted the importance of NK cells as regulatory cells that interact with dendritic cells, macrophages, T cells, and endothelial cells [39].

NK cell activation and lytic capacity is controlled by a balance between cell surface activation and inhibitory receptors. Activating receptors detect the presence of ligands, such as the stress-induced self-ligands that are recognized by NKG2D activating receptors [40]. The ideal cytokine environment provides the right interactions between NK cells and other immune cells, and cytotoxic and cytokine response of NK cells depends on this interaction. Type I IFN cytokines, such as IL-2, IL-12, and IL-15, are potent activators of NK cells [41]. Specifically, IL-2 can influence cytotoxic, proliferative, and cytokine producing activities of NK cells. One study found that GSH in combination with IL-2 and IL-12 augments NK cell functions and leads to control of *M. tb* infection [37]. Liposomal GSH supplementation restores redox homeostasis, and improves the immune response against *M. tb* infection, by further shifting the cytokine profile to a Th1 response [24]. Considering the antimycobacterial and immune-modulating properties of GSH through its action on NK cell activity and restoration of Th1 cytokine response, GSH in combination with Th1 cytokines, such as IL-2 and IL-12, enhances the host response against *M. tb* infection [24,42].

NK cells inhibit intracellular H37Rv growth in monocytes in vitro [37]. Furthermore, GSH supplementation along with cytokines augments NK cells’ function to completely inhibit H37Rv growth inside human monocytes [42]. In the same study, NK cells treated with N-acetyl cysteine (NAC), a GSH prodrug, in conjunction with cytokines, such as IL-2 and IL-12, had increased expression of cytotoxic ligands, FasL, and CD40L. NK cell cytotoxic receptors, NK activation receptors, and NK cytotoxic ligands on the cell surface of NK cells are correlated with the inhibition of *M. tb* growth [42]. In the end, NK cells require adequate levels of GSH and particular cytokines to function properly and to play their intended role in the innate immune response against tumor cells and parasitic infections, especially *M. tb* infection.

## 6. T Cells

Th1 cells are important for the control of *M. tb* infection. Their production of IL-2, IFN-γ, and TNF-α is necessary for the activation and cell recruitment of macrophages [14,43]. T cells are also a major component of granulomas and play a critical role in the immune response against *M. tb* infection [14]. Active TB is characterized by the suppression of this type of T cell response, as can be seen with the decreased levels of Il-2 and IFN-γ [44,45,46,47,48]. Furthermore, there is an overproduction of immunosuppressive cytokines IL-10 and TGF-β when T cell function is decreased during TB. Treatment of T cells derived from HIV patients with NAC results in reduction in the growth of *M. tb* inside monocytes [21]. GSH levels are lower in T cells isolated from HIV patients. However, NAC-treatment of T cells also resulted in an increase in the levels of GSH, which leads to enhanced functions of T lymphocytes in controlling *M. tb* infection [21].

Activated T cells produce reactive oxygen species (ROS), triggering antioxidative GSH responses [49]. Therefore, GSH is important for T cell effector functions. In a knock-out mice experiment, GSH lacking T cells undergo normal activation but cannot meet energy and biosynthetic requirements [49]. Activated T cells undergo clonal expansion, which raises the need for glucose and glutamine utilization, and increases the production of ROS [50,51,52,53,54]. T cell receptor activation activates mitochondrial ROS production, and prevention of lipid oxidation by glutathione peroxidase 4 (Gpx4) is important for survival and further activation [49]. Additionally, GSH-deficient T cells can undergo activation but cannot reprogram metabolism to meet their energy needs. GSH deficiency compromised activation of mTOR and expression of Myc transcription factor [49]. Myc is linked to glutaminolysis and glycolysis types of metabolism, and T cells with reduced Myc, switch to fatty-acid beta oxidation utilization [55,56]. In early proliferation, T cells require fatty acids for lipid and membrane synthesis. These components are provided by processes that increase acetyl-CoA synthesis from glucose and glutamine and fuels energy [49]. Without proper GSH levels, this process is disturbed, and if rising demand for acetyl-CoA levels are not met by the T cells, they eventually undergo necrosis.

## 7. GSH in HIV Patients

Studies on HIV-positive individuals have shown that GSH increases lymphocyte activation, which is crucial to the pathophysiology of HIV infection [57]. These studies have demonstrated that these individuals have low GSH levels. Decreased function of the immune system is correlated with HIV progression. Free-radical overload of monocytes and granulocytes is one-way immunological function declines. This results in a lack of antioxidant defenses, which could cause the CD4 cell loss that is frequently observed when HIV progresses [58]. According to further studies, when GSH levels are elevated, nuclear factor kB (NFkB), which is required for the HIV provirus’s active transcription, is activated [59,60,61]. Replenishing GSH is a potential therapeutic treatment in individuals with HIV since the primary pathophysiology of the HIV virus is the loss of CD4+ T cells.

In a previous study, our lab showed that treatment with 10 mM NAC, a GSH precursor, leads to reduction in the growth of *M. tb* inside monocytes from HIV patients compared to other treatment conditions, but did not show complete stasis of *M. tb* growth as was seen in healthy individuals [21]. T cells from healthy participants treated with NAC were shown to have higher levels of IL-12, IL-2, and IFN-γ. These cytokines are essential in controlling intracellular infections, such as *M. tb,* as they induce the Th1 response. Cell-mediated immunity is greatly diminished as a result of HIV’s high affinity for infecting and killing CD4+ T cells. This lowered immunity makes opportunistic infections, including *M. tb*, more frequent. GSH at low levels is important for cell-mediated immunity since it has been related to CD4+ T cell death, the main pathology of HIV infection [62,63,64,65]. Consequently, GSH may be an effective adjunctive treatment for those patients with HIV and *M. tb* infection [21].

GSH is produced by almost all cell types and as mentioned earlier, it comes in two forms: Reduced (rGSH) and oxidized (GSSG). The synthesis of rGSH has two distinct pathways (Figure 1). The two-step de novo production of rGSH requires the separate enzymes glutathione synthetase (GSS) and glutamate–cysteine ligase (GCL) (Figure 1). Additionally, GSSG is reduced by glutathione reductase (GSR) to create rGSH [60]. We have further conducted experiments to identify the reasons for decreased levels of GSH in HIV-infected people by comparing the extent to which the levels of GCLC, GSS, and GSR are lower in RBCs separated from HIV-infected patients compared to healthy participants. Levels of GSS and GCLC in the RBCs of HIV-infected individuals were found to be significantly lower than those of healthy individuals [11,21]. These studies have demonstrated that GSR expression is dramatically reduced in RBCs taken from HIV-positive subjects. This supports the cause of low GSH levels and the effects of GSH shortage, such as the decline in immunological function seen in HIV patients. Important research revealing lower levels of GSS, GCLC, and GSR in HIV-infected subjects confirms the notion and earlier findings that those with HIV-infections had lower amounts of GSH than healthy individuals [11].

The increased production of proinflammatory cytokines (Table 1), such as IL-1, IL-17, and TNF-α, has been hypothesized to be linked to chronic HIV infection. This rise in proinflammatory cytokine levels in the HIV-infected subjects correlated with an increase in free radical production. Free GSH is able to scavenge free radicals produced as a result of the proinflammatory cytokines’ chronic overproduction. The increased generation of free radicals in HIV-positive individuals will lead to the depletion of GSH [11]. Additionally, it was found that HIV-infected patients had significantly higher levels of TGF-β in their plasma and macrophage supernatants. This increase correlated with a decrease in the expression of the GCL gene in macrophages [11]. Moreover, increased TGF-β inhibits the formation of GCLC, which lowers the generation of new GSH molecules.

## 8. GSH in Type 2 Diabetes Patients

According to recent data, individuals with diabetes are two to three times more likely to develop TB than the general public. Diabetes is related to 10% of TB cases, and diabetic patients are more likely to die during TB therapy and experience relapse following treatment [66,67]. Another risk for diabetic patients is an increase in pro-inflammatory cytokines, such as IL-6 and IL-17, which cause chronic inflammation [12]. Furthermore, reduced expression of GSH synthesizing enzymes, such as GCLC, GSS, and GGT, is associated with lower GSH levels in people with type 2 diabetes mellitus (T2DM) [68].

There is evidence that individuals with T2DM have altered GSH synthesis and metabolism, as well as higher amounts of ROS and pro-inflammatory cytokines. Studies have shown that GSH concentrations in erythrocytes and plasma are lower in patients with T2DM, associated with reduced levels of the enzymes necessary for GSH production [69]. Depletion of GSH and excessive production of ROS cause an increase in the prevalence of TB-T2DM co-infection, which alters the immunological response and leads to worse outcomes in these individuals [70].

GSH has been studied as a possible TB adjunctive therapy in patients with T2DM. Liposomal GSH (lGSH) supplementation in T2DM individuals can lower overall oxidative stress, lower levels of oxidized GSH (GSSG), and maintain levels of reduced GSH in different blood components [71]. Additionally, lGSH supplementation helps patients with T2DM and was shown to increase the control of in vitro mycobacterial infections by reducing redox imbalance. lGSH combination therapy with everolimus, a mammalian target of rapamycin (mTOR) inhibitor, has also been studied as an adjunctive treatment for TB in T2DM patients. Everolimus is a small molecule that could modulate autophagy to enhance the killing of mycobacteria via inhibition of the mTOR pathway. In human studies, mTOR inhibition by everolimus has been shown to promote autophagy and improve the cellular innate immune response [72].

Another treatment option, a first-line antibiotic for *M. tb* infection, is rifampin (RIF). By interacting with and inhibiting mycobacterial DNA-dependent RNA polymerase, RIF prevents *M. tb* from being able to proliferate, but causes oxidative stress [73,74,75]. When RIF and lGSH are provided together in diabetic mouse models, malondialdehyde (MDA) measurements of oxidative stress are dramatically reduced, indicating a possibility for lGSH to counteract the negative oxidative effects of RIF while boosting its bactericidal benefits [74,75,76,77]. IL-6, a proinflammatory cytokine, is also thought to contribute to the development of oxidative stress. The levels of IL-6 in the lungs of diabetic mice treated with RIF and GSH and infected with *M. tb* were significantly decreased [78]. By reducing oxidative stress and boosting RIF bactericidal action, these findings highlight the value of lGSH as an adjunct therapy for *M. tb* infection. Levels of IL-12 and IFN-γ, cytokines important in controlling *M. tb* infections, were markedly elevated in the lungs of diabetic mice infected with *M. tb* across all treatment groups and were noticeably elevated further in animals treated with a combination of RIF and lGSH. lGSH can be added to RIF therapy to help in the control of the immune system, which will improve the ability to manage *M. tb* infection and lessen the risk of serious complications in diabetics [78]. The findings from lGSH trials on mice should encourage additional research into the potential of lGSH as an adjunctive therapy for *M. tb* infection in humans.

## 9. Clinical Trials Using GSH as Adjunct Therapy

TGF-β, a cytokine produced by regulatory T cells (T-Regs) and macrophages, is well known for its function in controlling immune responses by limiting the proliferation and expansion of T cells. The levels of GSH in the peripheral blood mononuclear cells (PBMCs) has been shown to be significantly decreased in HIV-infected individuals, which was linked to elevated TGF-β levels. At 13 weeks post-treatment, studies show a decline in TGF-β, taking into consideration how supplementing with GSH lowers levels of immunosuppressive cytokines [24]. Interestingly, liposomal glutathione (lGSH) supplementation for 13 weeks resulted in a significant decrease in the levels of IL-6 in HIV-positive individuals. In line with the diminished levels of GSH and increased levels of TGF-β, IL-6, and free radicals, we found decreased levels of Th1-specific cytokines, such as IL-2, IL-12, and IFN-γ in HIV-positive individuals (Table 1). Interestingly, 13 weeks of lGSH supplementation led to a significant drop in IL-6 levels in HIV-positive individuals [24].

Lower levels of Th1-specific cytokines, such as IL-2, IL-12, and IFN-γ, were detected in HIV-positive people, which is consistent with the decreased levels of GSH and higher levels of TGF-β, IL-6, and free radicals. These findings show a considerable rise in IFN-γ levels in HIV-infected individuals 13 weeks after lGSH administration. Further findings have demonstrated a significant rise in IL-1 and TNF-α levels in HIV-infected persons 13 weeks after lGSH intake. According to current data, there is no considerable change in the levels of IL-2 or IL-17 following lGSH administration. At 13 weeks, lGSH supplementation was successful in lowering IL-10 levels. Further results show that HIV disrupts the body’s physiological processes, causing an imbalance in the cytokine profiles. Supplementing with lGSH can restore immunological responses, which could be beneficial for HIV patients in treating opportunistic infections [24].

In a double-blinded study, this lab gave an empty liposomal supplement or an lGSH supplement to a group of HIV-infected people with CD4+ T cell counts < 350 cells/mm^3^ to take over a 3-month period. Measurements showed that IL-2, IL-12, and IFN-γ levels were significantly lower at baseline in HIV-positive patients compared to healthy individuals, but levels of IL-6, IL-10, TGF-β, and free radicals were significantly higher [79]. When HIV-positive participants were given lGSH for 3 months, their levels of IL-12, IL-2, and IFN-γ increased significantly in comparison to those of the placebo group, while their levels of IL-6, IL-10, and free radicals decreased, and their levels of TGF-β, IL-1, and IL-17 remained stable. While GSH levels rose in the treatment group, free radical levels in CD4+ T cells stabilized. For the course of the study, there was no discernible difference in the placebo group. In summary, supplementing with lGSH can assist in restoring redox homeostasis and cytokine balance in HIV-infected patients with CD4+ T cell counts below 350 cells/mm3, supporting the immune system’s ability to fight opportunistic infections, such as *M. tb* [79].

One of the features of TB is oxidative stress, which is evident by elevated lipid peroxidation products, such as malondialdehyde (MDA), as well as decreased antioxidant capacity [80,81]. TB patients are low in vitamins A, C, E, and also selenium and GSH [82,83]. NAC, a GSH precursor, was shown to limit *M. tb* infection by suppressing the oxidative stress induced by the infection and also having direct antimicrobial activity [84,85]. The direct effect of GSH involves enhancing nitric oxide (NO). When combined with GSH, NO forms S-nitrosoglutathione (GSNO) [81]. NAC is a synthetic form of cysteine that has been shown to lower ROS levels and improve the control of *M. tb* infection [25]. As a precursor to GSH, NAC can also prevent ferroptosis given that it can restore proper GSH levels to prevent oxidative stress. Notably, patients taking NAC showed a significant increase in GSH levels and improvement in antioxidative activity compared to the control group [84].

## 10. Conclusions

Adequate GSH levels and balanced levels of free radicals are important for an effective immune response against a vast number of infections. Specifically, TB is endemic and one of the leading causes of death in every part of the world. Through research and experiments, our growing understanding of TB and patients’ responses to treatments has improved and increased the number of available therapy options for TB patients. *M. tb* infection causes a cytokine imbalance and changes the balance of free radicals by causing oxidative stress. One molecule that has shown promising results for treatment of TB is GSH. GSH, in addition to other treatment options, such as Th1 cytokines, everolimus, and first-line antibiotics, such as rifampin, show significant improvements in the levels of GSH, free radicals, and cytokine profile. We have also highlighted the importance of GSH for patients with HIV or type 2 diabetes, who are more susceptible to TB. These individuals have increased pro-inflammatory cytokines in their bodies leading to more oxidative stress and generation of free radicals. Therefore, GSH can play a role in creating a balanced redox activity. GSH is a naturally synthesized molecule by our body’s cells that shifts the immune response toward a beneficial state and should be a part of an adjunctive therapy against TB.

## Figures and Tables

**Figure 1 biomedicines-11-01340-f001:**
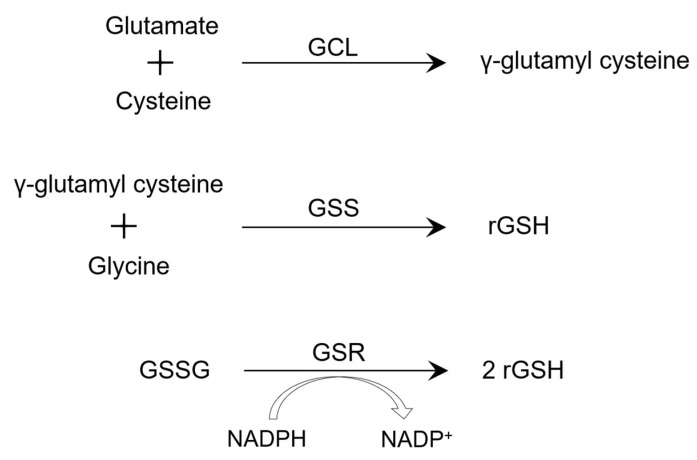
Thiol-reduced form of glutathione (rGSH) is synthesized through two different pathways. The first and rate-limiting step involves the formation of γ-glutamyl cysteine from the joining of glutamate and cysteine using glutamate–cysteine ligase (GCL) as the enzyme. The second step uses glutathione synthase (GSS) to join glycine to γ-glutamyl cysteine, to form rGSH. An alternative pathway involves reduction in oxidized GSH (GSSG), which is catalyzed by glutathione reductase (GSR) using NADPH as a cofactor, to reduce GSSG to two rGSH molecules.

**Table 1 biomedicines-11-01340-t001:** Types and functions of pro- and anti-inflammatory cytokines. Grouped into three categories: T helper 1, pro-inflammatory, and anti-inflammatory cytokines.

Type	Cytokine	Functions
T helper 1	IFN-γ	Activates macrophages and induces differentiation; stimulates NK cells and neutrophils
	TNF-α	Produced predominately by macrophages, induces necrosis or apoptosis
	IL-12	Induces production of IFN-γ and Th1 T cell response, and forms a link between innate and adaptive immune responses; activates NK cells
	IL-2	Enhances T cell viability and proliferation; enhances the generation of effector and memory cells; activates NK cells
Pro-inflammatory	IL-17	Cytokine that links T cell activation to neutrophil mobilization and activation
	IL-6	Elevated in chronic inflammation and high oxidative stress; recruitment of neutrophils and macrophages
Anti-inflammatory	IL-10	Immunosuppressive cytokine; inhibits IFN-γ and IL-12 production
	TGF-β	Immunosuppressive cytokine; inhibits T cell proliferation and function

## Data Availability

Not applicable.

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
