# Peer review of "Glutathione Modulates Efficacious Changes in the Immune Response against Tuberculosis"

_biomedicines, 2023, doi:10.3390/biomedicines11051340_

Round 1
Reviewer 1 Report
Comments to the Authors of manuscript number: biomedicines-2361281 entitled “Glutathione Modulates Efficacious Changes in the Immune Response Against Tuberculosis”.
1. Abstract is free from references. It should be corrected
2. “which is where it’s synthesized through two ATP-requiring enzymatic steps” - please make sure what was meant to be said here
3. The title of the review presented relates only to tuberculosis. However, even no short information is not found about tuberculosis in humans or animals. On the other hand, such organization of the manuscript suggests that the two other parts relating to HIV patients and the role of GSH in type 2 diabetes mellites is not linked with TB.
4. the part about TB should be added, where Authors could explain the relationships between TB and HIV and DM.
5. The part about searching literature should be added, as it is generally practiced in reviews. What words were used to collect information. what was found, excluded and why, and what decided that any paper was included. figure would be helpful.
Author Response
Reviewer 1
Comment 1: Abstract is free from references. It should be corrected.
Response 1: We thank the reviewer for this comment. We would gladly incorporate this suggestion; however, after checking with the template and instructions, and previous publications in the journal, it seems that it is standard not to include any references in the abstract.
Comment 2: “which is where it’s synthesized through two ATP-requiring enzymatic steps” - please make sure what was meant to be said here.
Response 2: Thank you for bringing this to our attention. We have rephrased the sentence to “Glutathione (GSH) is a tripeptide and an antioxidant, which is mostly found in the cytosol, and synthesized through two ATP-requiring enzymatic steps.”
Comment 3: The title of the review presented relates only to tuberculosis. However, even no short information is not found about tuberculosis in humans or animals. On the other hand, such organization of the manuscript suggests that the two other parts relating to HIV patients and the role of GSH in type 2 diabetes mellitus is not linked with TB.
Response 3: Thank you for this comment. We have added a section about TB in the Introduction and added details regarding the two conditions. We have made sure each section talks about GSH and if appropriate about TB. Not every section seemed to be relevant to TB but most if not all sections have information regarding GSH.
Comment 4: the part about TB should be added, where Authors could explain the relationships between TB and HIV and DM.
Response 4: Thank you for this suggestion. We have added a section about information on TB and also added details about how the two conditions can make individuals more susceptible to TB.
Comment 5: The part about searching literature should be added, as it is generally practiced in reviews. What words were used to collect information. what was found, excluded and why, and what decided that any paper was included. figure would be helpful.
Response 5: We thank the reviewer for this comment. We have added a section on methodology, and how we collected the information. We have added what we were looking for, and what types of articles were excluded from our review. We have also added the phrases and terminology that were used in the search engines to find articles.

Reviewer 2 Report
Thank you very much for allowing me to review the article entitled "Glutathione Modulates Efficacious Changes in the Immune Response Against Tuberculosis" (biomedicines-2361281), which is submitted to the Section "Molecular and Translational Medicine" in the Special Issue "Oxidative Stress and Inflammation: From Mechanisms to Therapeutic Approaches."
The objective of this review is to analyze the role of GSH supplementation in forming a proper immune response to M. tuberculosis and its use as an adjunctive therapy for tuberculosis (TB) patients. The authors have based their review on their laboratory experience.
The summary should follow a structured approach, including the objective, methodology, and conclusion of the work. It is essential to include the methodology used to carry out this review since it is crucial to know the databases that were searched, and the period of time reviewed, in order to connect with other review studies in the future.
In the conclusion, it is appropriate to state the agreements and disagreements found in the reviewed literature. However, it is not appropriate to speak of a "demonstration."
Some of the cited literature is outdated, dating back to 1979 and 1988. As this is a review study, I suggest that the authors replace this older literature with more current sources.
Author Response
Reviewer 2
Comment 1: The summary should follow a structured approach, including the objective, methodology, and conclusion of the work. It is essential to include the methodology used to carry out this review since it is crucial to know the databases that were searched, and the period of time reviewed, in order to connect with other review studies in the future.
Response 1: We thank the reviewer for this comment. We have added a section on methodology. We have provided the terms and phrases that were used on the search engines to find the articles. We have added the criteria that were used to find the articles and what made us exclude an article.
Comment 2: In the conclusion, it is appropriate to state the agreements and disagreements found in the reviewed literature. However, it is not appropriate to speak of a "demonstration."
Response 2: We thank the reviewer for this comment. We have removed the phrase “demonstration” and reworded that sentence. Also added more information in the conclusion to make it more relevant and appropriate to the review.
Comment 3: Some of the cited literature is outdated, dating back to 1979 and 1988. As this is a review study, I suggest that the authors replace this older literature with more current sources.
Response 3: We thank the reviewer for this comment. We have addressed this issue and replaced multiple citations throughout the review article with as many recent sources as possible. We have also added more recent references to make the review study as current as possible.

Reviewer 3 Report
This review-paper I carefully reviewed, the Authors aimed to analyze the role of GSH supplementation in forming proper immune response to M. tb and when used in adjunctive therapy for tuberculosis patients.
The topic reviewed and investigated is of significant interest for human health.
A good amount of literature was collected and discussed.
The paper organization and structure are quite precise.
The findings have been properly reported and the available data well discussed by using the updated literature including recent findings by other authors.
The conclusion section provides a clear overview of the findings and their usefulness, evel if it could be further improved.
As specific comments, in order to further improve the quality of the paper, I suggest to:
- Try to further improve the Abstract section;
- The English language is fine;
-Some additional recently published references may add value to the Introduction section;
- Check all acronyms used, spell at first use;
- The references have been reported in an appropriate form and edited according to the journal's guidelines.
So, based on my opinion, I think that this paper merits the acceptance after very minor revision.
Author Response
Reviewer 3
Comment 1: Try to further improve the Abstract section.
Response 1: We thank the reviewer for this suggestion. We have further improved it and added information to the abstract to make it more relevant to the article.
Comment 2: Some additional recently published references may add value to the Introduction section.
Response 2: We thank the reviewer for this comment. We have added a section about TB and information about the disease. We have added more articles to that section and also as many recent articles as possible.

Round 2
Reviewer 1 Report
I have no more comments
Reviewer 2 Report
I have carefully reviewed the new version submitted by the authors of the article entitled "Glutathione Modulates Efficacious Changes in the Immune Response Against Tuberculosis" (biomedicines-2361281), which has been submitted to the Section "Molecular and Translational Medicine" in the Special Issue "Oxidative Stress and Inflammation: From Mechanisms to Therapeutic Approaches". I have also reviewed the response to the comments made.
The presented review article has clarified all the aspects that were identified, and therefore, I consider that this review collates reports that demonstrate the benefits of GSH in improving immune cell responses against M. tb infection, as well as the use of GSH as an adjunctive therapy for TB.